# Perioperative Neurocognitive Disorder in Individuals with a History of Traumatic Brain Injury: Protocol for a Systematic Review and Meta-Analysis

**DOI:** 10.3390/biology14020197

**Published:** 2025-02-13

**Authors:** Zeeshan Ahmad Khan, Dewan Md. Sumsuzzman, Tahiris A. Duran, Ling-Sha Ju, Christoph N. Seubert, Anatoly E. Martynyuk

**Affiliations:** 1Department of Anesthesiology, College of Medicine, University of Florida, JHMHC, 1600 SW Archer Road, Gainesville, FL 32610-0254, USA; zkhan2@anest.ufl.edu (Z.A.K.); durantahiris@ufl.edu (T.A.D.); lju@anest.ufl.edu (L.-S.J.); cseubert@anest.ufl.edu (C.N.S.); 2Agent-Based Modelling Laboratory, York University, Toronto, ON M3J 1P3, Canada; dewanp@yorku.ca; 3McKnight Brain Institute, College of Medicine, University of Florida, Gainesville, FL 32611, USA

**Keywords:** general anesthetics, inflammation, meta-analysis, perioperative neurocognitive disorder, traumatic brain injury

## Abstract

Preexisting neurodegenerative changes, stress, and inflammation that become more common with advanced age may contribute to accelerated neurocognitive decline after general anesthesia (GA) and surgery, a condition known as postoperative neurocognitive disorder (PND). PND affects millions of senior patients every year. Whether specific groups of young adults with pathophysiological conditions that involve dysregulated stress response systems, neuroinflammation, neurodegenerative, and neurocognitive abnormalities, such as traumatic brain injury (TBI), are also more susceptible to PND is not known. We will analyze published research data to determine whether patients with a history of mild to moderate TBI have an increased risk of developing PND after surgery under GA. We will use several medical databases and both manual and AI tools for data extraction and quality assessment. The results of the proposed review will help to improve clinical practices and suggest new research directions. This protocol paper for the proposed review is crucial as it outlines the methods and criteria, ensuring transparency, reproducibility, and reliability of the findings.

## 1. Introduction

Perioperative neurocognitive disorder was recently introduced as an overarching term for cognitive deficits diagnosed in the preoperative and/or postoperative period [1]. Postoperative neurocognitive abnormalities include acute deficits that last from up to a week (postoperative delirium) to a month (delayed neurocognitive recovery) post-GA/surgery. The acute symptoms present as confusion, agitation, delusions, and deficiencies in orientation and attention. The declines in memory, awareness, reasoning, judgment, and language that can be determined by neuropsychological tests during a period up to 12 months after GA/surgery are referred as a postoperative neurocognitive disorder (PND) [1]. PND is an important public health problem potentially affecting millions of patients every year [2]. Although the exact etiology of PND remains poorly understood, clinical and laboratory evidence suggests that accompanying stress, neuroinflammation, and preexisting neurodegenerative diseases affecting memory and other cognitive functions play an essential role in PND development [3,4]. Because neurodegenerative diseases worsen with age, clinical and laboratory studies of PND primarily focus on aging population [5]. However, the actual number of patients vulnerable to PND may be higher. Thus, patients within a wide range of ages suffer from pathophysiological conditions that are accompanied by neuroinflammation and dysregulated stress response systems [6,7]. Such conditions may predispose patients to neurological/neurocognitive diseases and, by extension, to the development of PND after undergoing GA/surgery, regardless of their age. One such condition is traumatic brain injury (TBI) [8].

With over 50 million cases per year, traumatic brain injury (TBI) affects patients of all ages, including younger adults—especially those involved in contact sports and military service [9]. Patients with TBI often require general anesthesia (GA)/surgery for the treatment of accompanying non-brain-related conditions, such as orthopedic, abdominal, or thoracic injuries (extracranial surgeries) [10]. Many of these patients become parents after TBI/GA/surgery. If the inflammatory and stress effects of TBI/GA/surgery interact to induce more profound outcomes, not only TBI/GA/surgery-exposed patients but also their future unexposed offspring may be at risk of developing neurocognitive abnormalities. Such a possibility is supported by our two recent rat studies, which are the only known to us animal studies investigating the heritable effects of parental TBI. Using a rat model of TBI, we found that TBI in young adult male rats leads to neurocognitive deficits in their offspring that are more profound in male offspring [11]. Furthermore, in direct relevance to the topic of this study, we observed the interactions of the effects of surgery, TBI, and subsequent repeated anesthesia with sevoflurane in inducing neurocognitive abnormalities in both generations [12]. Considering that in rats the contribution of sires to offspring development is limited to conception, these findings support a biological (before birth) mode of intergenerational transmission of paternal TBI effects to offspring [11,12]. Importantly, epidemiological studies find that children of parents with TBI are more likely to develop psychiatric disorders [13,14,15,16]. Although the mechanisms of intergenerational transmission of such abnormalities in humans remain unresolved, similarity with observations of animal studies that abnormalities were more profound in sons [11,12,15] suggests a possibility of a biological mode of intergenerational transmission of parental TBI effects in humans as well—occurring before offspring birth. Altogether, these limited animal and human observations highlight the importance of filling the gap in understanding the neurocognitive outcomes not only in patients with TBI/GA/surgery but also the neurocognitive consequences for their children.

Therefore, this study aims to systematically review and meta-analyze published studies to address the question of whether TBI and extracranial surgery under GA interact to induce neurocognitive abnormalities, or PND, in the exposed patients and their children. By addressing this question, the study seeks to enhance our understanding of patients’ vulnerability to PND and inform strategies to mitigate cognitive risks associated with TBI and surgeries under GA.

## 2. Experimental Design

### 2.1. Data Sources

The following databases will be used for retrieving articles: Ovid Medline, Embase, Ovid Emcare, Global Health, and APA PsycInfo. The search strategy will be to use the “advanced” search section with the AND/OR function and use relevant keywords, such as TBI, anesthesia, and extracranial surgery. No restrictions on the year of publication and language will be imposed. The reference list of eligible records that meet our inclusion criteria will be manually searched. The full search strategy for all the databases is provided in Appendix A.

### 2.2. Study Design

#### 2.2.1. Eligibility Criteria

Observational studies, such as cohort, case–control, longitudinal, and cross-sectional designs, will be used in this meta-analysis to examine PND in patients with TBI and extracranial surgeries under GA. Reviews and meta-analyses will be excluded, as they do not provide new primary data.

#### 2.2.2. Participants

Our participant pool will consist of patients with a history of mild to moderate TBI and extracranial surgery under GA, as well as, if data exist, their offspring. The severity of TBI is assessed using the Glasgow Coma Scale (GCS). We will include articles that focus on mild (GCS 13–15) or moderate (GCS 9–12) TBIs (average GCS ≤ 12) [17]. This study will exclude patients with severe TBI (GCS < 8), as they often are required to go through intracranial surgeries [18]. Any non-clinical or animal study will also be excluded.

#### 2.2.3. Intervention

This study will only include studies where the patients have mild to moderate TBI and extracranial surgery under GA, and their offspring, if relevant data are available. This study excludes intracranial surgeries, as they can be a confounding variable that directly adds neurological morbidity [19]. This study will also exclude cardiac and pulmonary bypass surgeries as these can also add further brain damage outside of TBI itself [20,21].

#### 2.2.4. Control

This meta-analysis will include patients with a history of mild to moderate TBI who have not undergone any kind of general anesthesia/surgery and their offspring. Those with severe TBI will continue to be excluded.

## 3. Outcomes

The primary outcomes are long-term neurocognitive outcomes, such as deficiencies in memory, awareness, reasoning, judgment, and language (postoperative neurocognitive disorder). This includes neurocognitive markers that are often associated with TBI studies, such as the trail-making test, the Wechsler Adult Intelligence Scale—Fourth Edition Processing Speed Index, and the Rey Auditory Verbal Learning Test [22,23,24]. To explore the role of stress and inflammation in PND, we will analyze key inflammatory markers, such as IL-1, IL-6, IL-10, TNF-α, and CRP, as well as the stress hormone cortisol [25]. This systematic review and meta-analysis will also include epigenetic outcomes regarding any intergenerational effects from a parent with a history of TBI and extracranial surgery under GA to their offspring.

## 4. Detailed Procedure

### 4.1. Study Selection

At least two independent reviewers (Z.A.K. and D.T.) will screen and select studies in which they must agree on articles to be used in the final analysis. In case of disagreements, other reviewers (L.S.J. and D.M.S.) will be consulted to select/reject the study. Data will be collected manually after screening and deleting duplicates initially using Rayyan software [26]. A few agreed-upon articles will be selected as relevant, and the deduplicated dataset will be put through an artificial intelligence screening process in ASReview to continue filtering more relevant articles [27,28]. Two independent researchers will share data using the ASReview software (Version 1.6.2). Since the ASReview software can only be used by one person at a time, the researchers will switch access between each other during the screening process. Researcher 1 (Z.A.K.) will screen 400 articles before switching access to the other researcher (D.T.) until completion of the screening process. Combined, the researchers will attempt to screen 20% of all data (*n* = 19,124), but the screening will be concluded after 500 articles have been screened since the last relevant paper. The relevant outcome measures will be compiled from each article and will be utilized for data analysis. Studies that are selected will be stored in a Dr. Martynyuk laboratory shared drive and LabArchives at the University of Florida to manage articles.

### 4.2. Data Extraction

We will collect data such as first authors, year of publication, patient age, gender, BMI, time of anesthesia, type of anesthesia, duration of anesthesia, race, education, income, health insurance status, country, duration of post-operative stay, time of cognitive assessment after surgery and anesthesia, return to work, comorbidity, diagnosis, and exposure. GetData Graph Digitizer will be used to extract data that are presented in the form of graphs in the articles [29]. Prior to the actual data extraction stage, we plan to pilot at least five articles. While we may initially assume certain variables to extract from the selected articles, the practical scenario may differ. Therefore, piloting these articles will be crucial for adjusting our initial plans based on the findings.

### 4.3. Quality Assessment

The Newcastle–Ottawa Scale will be used to evaluate the risk of bias in the observational studies included in our meta-analysis [30]. This tool assesses studies across three domains: selection (whether cases and controls or cohorts are representative and appropriately identified), comparability (how well studies account for confounding factors), and outcome assessment (clarity and reliability of outcome measures). Studies will be scored in each domain, with higher scores indicating a lower risk of bias. This evaluation will help categorize studies as low, moderate, or high risk of bias, ensuring that our conclusions are based on the most reliable evidence and addressing any potential limitations [30].

### 4.4. Data Synthesis and Analysis

In our meta-analysis, we will rigorously address various types of data (continuous and binary outcome) using appropriate statistical methods. For continuous outcomes, we will apply the mean difference using random effect modelling when studies measure outcomes using the same scale, ensuring a direct comparison [31]. When studies use different scales, we will utilize the standardized mean difference to standardize the data, allowing for accurate pooling of results across diverse measurement tools [31,32]. Additionally, we will consider the use of a fixed effects model when the population is homogeneous. The effect size will be interpreted using Cohen’s criteria: small (0.2), moderate (0.5), or large (0.8) [33]. For binary outcomes, we will calculate the odds ratio (OR) for case–control studies and the risk ratio (RR) for cohort studies, which will enable us to assess the effect size across different studies [34]. When analyzing single-group binary data, we will perform proportional analysis by converting raw event counts into proportions before pooling, which will help standardize the data [35]. For the analysis of binary outcomes, we will employ arcsine-based transformations using packages such as metaprop packages in Stata/SE [36]. This approach will allow us to properly handle proportions and ensure the assumptions of the statistical models are met.

The review will incorporate the I^2^ statistic to gauge heterogeneity and ensure that our conclusions are based on a comprehensive and accurate synthesis of the data [37]. The I^2^ statistic will be incorporated with interpretations as follows: 0% to 25% indicates low heterogeneity, 26% to 50% suggests moderate heterogeneity, and above 50% reflects high heterogeneity [37]. Subgroup analysis will include the mechanism of anesthetics (e.g., GABAergic anesthetics, NMDA receptor antagonists, etc.), time and duration of anesthetic induction following TBI, type of surgery, sex, post-operative stay, race, education, income, and health insurance. We acknowledge that not all studies may report these variables, particularly socioeconomic factors and healthcare access (such as health insurance coverage). If such data are available, we will incorporate them in subgroup analyses or meta-regression models to explore their potential effects. In cases where data are insufficient or inconsistently reported, we will highlight this limitation and discuss its implications in the final manuscript. For numeric variables such as age, we will perform meta-regression. A minimum of five studies in each class will be required to perform subgroup analysis.

To ensure the robustness of our results, subgroup and sensitivity analyses will involve excluding one study at a time, removing high-risk studies, and examining the impact of publication years and countries. Meta-regression will be used to assess the effects of median age and cognitive/functional outcomes on the overall results. To minimize publication bias, we will employ funnel plots [38] and the Egger regression test [39] to detect and address potential biases. All statistical analyses will be performed using Stata/SE (version 16) [40], with a significance threshold set at a *p* value of <0.05. Continuous outcomes will be reported with mean difference/standardized mean difference and 95% confidence intervals. If quantitative synthesis is not appropriate, a qualitative synthesis can be planned, such as a scoping review.

## 5. Expected Results and Discussion

Our study aims to provide a comprehensive analysis of neurocognitive function in patients with a history of TBI who undergone extracranial surgeries under GA, and in their offspring, provided sufficient data on neurocognitive outcomes in offspring of parents with a history of TBI/GA/surgery exist. By systematically reviewing and meta-analyzing observational studies including cohort, case–control, longitudinal, and cross-sectional designs, we will address several critical knowledge gaps. The primary knowledge gap to be addressed is the possibility of a synergistic impact of TBI, extracranial surgery, and GAs on neurocognitive decline, which have been underexplored in existing research. Another aspect of this research will include assessing the roles of dysregulated stress response systems and inflammation in mediating cognitive abnormalities in TBI patients who underwent surgery under GA as well as their offspring.

The proposed meta-analysis will include patients with mild to moderate TBI who have undergone extracranial surgery under GA and patients with mild to moderate TBI who did not undergo any surgery under GA. The subjects with TBI will not include patients with severe TBI cases, as the intracranial complications and long-term neurological complications, generally associated with severe TBI, may interfere with detecting the interaction of the effects of TBI and surgery under GA that affect neurocognitive function, i.e., to induce PND [41].

We will explore the data by subgroup analyses and evaluate how various factors are associated with the studied population and intervention. We will examine how age affects cognitive decline, as well as the impact of GAs with different mechanisms of actions. Additionally, we will analyze the effects of various types of surgeries and differences in outcomes between genders. We will also evaluate how the timing of anesthesia/surgery relative to TBI influences neurocognitive function. To ensure the robustness of these subgroup analyses, a minimum of five studies will be required for each subgroup, providing sufficient data to detect statistically significant effects [42]. This approach will help us identify and understand the nuances in how different variables affect neurocognitive outcomes in patients with a history of TBI and surgery under GA, ultimately offering more tailored insights into patient management and treatment strategies.

As there is no previous systematic review and meta-analysis on post-operative neurocognitive decline following TBI and extracranial surgery with GAs, we will integrate data from five diverse databases to offer a more holistic view of the postoperative risks associated with TBI and extracranial surgeries under GA. Our approach, which includes rigorous sensitivity and meta-regression, will provide a detailed understanding of how different factors, such as age and cognitive assessments, influence outcomes. This thorough methodology combined with the use of various statistical tools, such as funnel plots and the Egger regression test to address publication bias, will strengthen the reliability of our findings.

The results of our study may contribute to an improvement in patients’ care by identifying factors that predispose patients with a history of TBI and extracranial surgery and their unexposed offspring to PND. These results may offer insights for the optimization of anesthesia regimens, timing of surgeries, and perioperative care protocols to minimize the risk of PND. The proposed study may also help to identify neurocognitive and molecular biomarkers of PND in patients with a history of TBI and their offspring and potential translational pharmacotherapeutic approaches to minimize the consequences of TBI. Additionally, by exploring intergenerational effects, our study may open new avenues for research in the field of neurocognitive complications associated with TBI, surgery, and GA.

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
