# Peer review of "Perioperative Neurocognitive Disorder in Individuals with a History of Traumatic Brain Injury: Protocol for a Systematic Review and Meta-Analysis"

_biology, 2025, doi:10.3390/biology14020197_

Round 1

Reviewer 1 Report

Comments and Suggestions for Authors

The introduction describes a concise proposal for a comprehensive meta-analysis exploring the available data among patients with a history of TBI and their risk of developing perioperative neurocognitive disorder (PND) after undergoing extracranial surgeries while under general anesthesia. Within the introductory section, three of the authors cite a previous animal-based model they had constructed on the behavioral disturbances observed in offspring with paternal TBI history and repeated exposure to sevoflurane. If this is the only animal-based model available in the literature, please declare this as a limitation. If it is not, please include additional animal-based studies and the observations made in the offspring of those afflicted with TBI and exposure to GA. Were there other anesthetic agents used besides sevoflurane?

Readers will benefit from the epidemiologic data extracted from studies conducted among children of parents with a history of TBI and exposure to GA in the setting of extracranial surgeries.  More specifically, we hope that a theorized mechanism will be proposed, based on the available data, as to how trauma may influence the development of psychiatric comorbidities in their offspring. It will remain crucial to stratify neurocognitive outcomes among those afflicted with TBI, as well as their offspring, based upon the specific extracranial operation performed under GA as this may influence clinicians' decision for surgical candidacy.  Likewise, neurocognitive outcomes must be stratified by which anesthetic agent was selected as this may influence the perioperative approach to anesthetic selection of available alternatives to help mitigate the risk of PND. 

If available, scales of quality of life among those afflicted by TBI/GA/extracranial surgery exposure -- stratified by the anesthetic agent used, and by the type of surgery performed -- could also influence medical decision-making in the pre and/or perioperative period. This will provide additional utility for the clinician to optimize alternative anesthetic agents for necessary extracranial surgeries. Long-term scales that quantify the quality of life among patients with a history of TBI who had previously undergone necessary extracranial surgery, requiring the use of GA, may guide formal goals of care discussions with patients and/or their surrogate decision-makers in the emergent and nonemergent settings. If this has yet to be explored in the available literature, please declare long-term outcomes on quality of life as an area for further exploration. 

Author Response

Comment #1: The introduction describes a concise proposal for a comprehensive meta-analysis exploring the available data among patients with a history of TBI and their risk of developing perioperative neurocognitive disorder (PND) after undergoing extracranial surgeries while under general anesthesia. Within the introductory section, three of the authors cite a previous animal-based model they had constructed on the behavioral disturbances observed in offspring with paternal TBI history and repeated exposure to sevoflurane. If this is the only animal-based model available in the literature, please declare this as a limitation. If it is not, please include additional animal-based studies and the observations made in the offspring of those afflicted with TBI and exposure to GA. Were there other anesthetic agents used besides sevoflurane?

Response: In the edited text, we explicitly state that the referenced animal studies are the only known studies to us on heritable effects of TBI and TBI pus surgery and/or anesthesia, and that the limited animal and human observations highlight the importance of filling the gap in understanding the neurocognitive outcomes not only in patients with TBI/GA/surgery but also the neurocognitive consequences for their children.

Lines 71-91: If the inflammatory and stress effects of TBI/GA/surgery interact to induce more pro-found outcomes, not only TBI/GA/surgery-exposed patients but also their future unexposed offspring may be at risk of developing neurocognitive abnormalities. Such a possibility is supported by our two recent rat studies, which are the only known to us animal studies investigating the heritable effects of parental TBI. Using a rat model of TBI, we found that TBI in young adult male rats leads to neurocognitive deficits in their offspring that are more profound in male offspring [11]. Furthermore, in direct relevance to the topic of this study, we observed the interactions of the effects of surgery, TBI, and subsequent repeated anesthesia with sevoflurane, in inducing neurocognitive abnormalities in both generations [12]. Considering that in rats the contribution of sires to offspring development is limited to conception, these findings support a biological (before birth) mode of intergenerational transmission of paternal TBI effects to offspring [11,12]. Importantly, epidemiological studies find that children of parents with TBI are more likely to develop psychiatric disorders [13–16]. Although the mechanisms of intergenerational transmission of such abnormalities in humans remain unresolved, similarity with observations of animal studies that abnormalities were more profound in sons [11,12,15] suggests a possibility of a biological mode of intergenerational transmission of parental TBI effects in humans as well—occurring before offspring birth. Altogether, these limited animal and human observations highlight the importance of filling the gap in understanding the neurocognitive outcomes not only in patients with TBI/GA/surgery but also the neurocognitive consequences for their children.”

Comment #2: Readers will benefit from the epidemiologic data extracted from studies conducted among children of parents with a history of TBI and exposure to GA in the setting of extracranial surgeries.  More specifically, we hope that a theorized mechanism will be proposed, based on the available data, as to how trauma may influence the development of psychiatric comorbidities in their offspring. It will remain crucial to stratify neurocognitive outcomes among those afflicted with TBI, as well as their offspring, based upon the specific extracranial operation performed under GA as this may influence clinicians' decision for surgical candidacy.  Likewise, neurocognitive outcomes must be stratified by which anesthetic agent was selected as this may influence the perioperative approach to anesthetic selection of available alternatives to help mitigate the risk of PND. 

Response: These are all important points. We will analyze the roles of all these and other factors, provided sufficient literature data is available. This plan is presented in the following paragraphs:

Lines 162-166: We will collect data such as.. patient and offspring age, gender, BMI, type of extracranial surgery, type and duration of anesthesia, duration of post-operative stay, return to work, comorbidity, diagnosis, time of cognitive assessment after surgery and anesthesia, race, education, income, health insurance status, and country.

Lines 202-210: “Subgroup analysis will include the mechanism of anesthetics (e.g., GABAergic anesthetics, NMDA receptor antagonists,..), time and duration of anesthetic induction following TBI, type of surgery, sex, post-operative stay, race, education, income, and health insurance. We acknowledge that not all studies may report these variables, particularly socioeconomic factors and healthcare access (such as health insurance coverage). If such data are available, we will incorporate them in subgroup analyses or meta-regression models to explore their potential effects. In cases where data are insufficient or inconsistently reported, we will highlight this limitation and discuss its implications in the final manuscript.”

Lines 241-251: We will explore the data by subgroup analyses and evaluate how various factors are associated with the studied population and intervention. We will examine how age affects cognitive decline, as well as the impact of GAs with different mechanisms of actions. Additionally, we will analyze the effects of various types of surgeries and differences in outcomes between genders. We will also evaluate how the timing of anesthesia/surgery relative to the TBI influences neurocognitive function. To ensure the robustness of these subgroup analyses, a minimum of 5 studies will be required for each subgroup, providing sufficient data to detect statistically significant effects [41]. This approach will help us identify and understand the nuances in how different variables affect neurocognitive outcomes in patients with a history of TBI and surgery under GA, ultimately offering more tailored insights into patient management and treatment strategies.”  

We will collect the following data to explore whether upregulated stress and inflammatory signaling is involved in PND in patients with a history of TBI/surgery/GA and their offspring

Lines 136-141: To explore the role of stress and inflammation in PND, we will analyze key inflammatory markers such as IL-1, IL-6, IL-10, TNF-α, and CRP, as well as the stress hormone cortisol [25]. This systematic review and meta-analysis will also include epigenetic outcomes regarding any intergenerational effects from a parent with a history of TBI and extracranial surgery under GA to their offspring.

Comment #3: If available, scales of quality of life among those afflicted by TBI/GA/extracranial surgery exposure -- stratified by the anesthetic agent used, and by the type of surgery performed -- could also influence medical decision-making in the pre and/or perioperative period. This will provide additional utility for the clinician to optimize alternative anesthetic agents for necessary extracranial surgeries. Long-term scales that quantify the quality of life among patients with a history of TBI who had previously undergone necessary extracranial surgery, requiring the use of GA, may guide formal goals of care discussions with patients and/or their surrogate decision-makers in the emergent and nonemergent settings. If this has yet to be explored in the available literature, please declare long-term outcomes on quality of life as an area for further exploration. 

Response: We agree that including the quality of life (QoL) among those afflicted by TBI, GA, and extracranial surgery exposure is crucial. We are considering ability to return to work (RTW) as a measure of QoL, as it is a frequently reported outcome in TBI research. Returning to work is often a significant indicator of QoL, recovery, and overall well-being for individuals with TBI. Studies have shown that successful RTW can reduce financial burdens, provide a sense of productivity, and improve overall QoL [2]. We will incorporate data on RTW as a QoL measure, stratified by the anesthetic agent used and the type of surgery performed.

  1. Figueredo JM, García-Ael C, Gragnano A, Topa G. Well-Being at Work after Return to Work (RTW): A Systematic Review. Int J Environ Res Public Health. 2020 Oct 15;17(20):7490. doi: 10.3390/ijerph17207490. PMID: 33076302; PMCID: PMC7602369.

Line 162-166: “We will also collect data such as first authors, year of publication, patient age, gender, BMI, time of anesthesia, type of anesthesia, duration of anesthesia, race, education, income, and health insurance status, country, duration of post-operative stay, time of cognitive assessment after surgery and anesthesia, return to work, comorbidity, diagnosis, and exposure.”

Line 202-205: Subgroup analysis will include the mechanism of anesthetics (e.g., GABAergic anesthetics, NMDA receptor antagonists,..), time and duration of anesthetic induction following TBI, type of surgery, sex, post-operative stay, race, education, income, and health insurance.”

Reviewer 2 Report

Comments and Suggestions for Authors

The authors aimed to analyze published research data to determine whether patients with a history of mild to moderate TBI have an increased risk of developing PND after surgery under GA.

It is a good attempt by the authors to choose this topic.

Do the authors wish to include case reports?

Will articles other than English be considered?

Do the authors aim to discuss the biomarkers?

Are the authors planning to include the post-operative stay?

Will race, lower education, lower income, and health insurance be included?

Author Response

Comment #1: The authors aimed to analyze published research data to determine whether patients with a history of mild to moderate TBI have an increased risk of developing PND after surgery under GA. It is a good attempt by the authors to choose this topic.

Response:  Thank you for your positive feedback. We appreciate your recognition of the importance of this topic.

Comment #2: Do the authors wish to include case reports?

Response: We have outlined our inclusion criteria to focus on observational studies, such as cohort, case-control, longitudinal, and cross-sectional designs, to examine PND in patients with TBI and extracranial surgeries under general anesthesia (GA). Reviews and meta-analyses are excluded as they do not provide new primary data.
To maintain the robustness and generalizability of our meta-analysis, we will exclude case reports from our study selection. Case reports, while valuable for providing detailed descriptions of individual patient experiences, do not offer the comprehensive data required for statistical analysis and drawing broader conclusions about trends and associations in larger populations. Including only observational studies such as cohort, longitudinal, and cross-sectional designs ensures that our analysis is based on primary data that reflect systematic investigation and provide more reliable and generalizable insights into PND in patients with a history of TBI undergoing extracranial surgeries under general anesthesia.

Comment #3: Will articles other than English be considered?

Response: Yes, we will include literature in languages other than English. We are searching five databases such as Ovid Medline, Embase, Ovid Emcare, Global Health, and APA PsycInfo. We are not putting any language filter/bar.

Line 104-105: “No restrictions on the year of publication and language will be imposed.”

Comment #4: Do the authors aim to discuss the biomarkers?

Response: We are collecting data of epigenetic changes, stress, and inflammation and we will discuss the biomarkers.

Line 136-141: To explore the role of stress and inflammation in PND, we will analyze key inflammatory markers such as IL-1, IL-6, IL-10, TNF-α, and CRP, as well as the stress hormone cortisol [24]. This systematic review and meta-analysis will also include epigenetic outcomes regarding any intergenerational effects from a parent with a history of TBI and extracranial surgery under GA to their offspring.

Comment #5: Are the authors planning to include the post-operative stay?

Response: Thank you for the insightful suggestion. In response to your recommendation, we have added the duration of post-operative stay to our subgroup analysis. This inclusion will enhance our understanding of its impact on patient outcomes. The updated subgroup analysis now includes variables such as the mechanism of anesthetics (e.g., GABAergic anesthetics, NMDA receptor antagonists), type of surgery, sex, route of anesthetics (such as intravenous, volatile, etc.), duration of anesthetic induction, the time between TBI and anesthetic induction, and the duration of post-operative stay.

Line 162-166: We will collect data such as first authors, year of publication, patient age, gender, BMI, time of anesthesia, type of anesthesia, duration of anesthesia, race, education, income, and health insurance status, country, duration of post-operative stay, time of cognitive assessment after surgery and anesthesia, return to work, comorbidity, diagnosis, and exposure.”

Line 202-205: Subgroup analysis will include the mechanism of anesthetics (e.g., GABAergic anesthetics, NMDA receptor antagonists,..), time and duration of anesthetic induction following TBI, type of surgery, sex, post-operative stay, race, education, income, and health insurance.

Comment #6: Will race, lower education, lower income, and health insurance be included?

Response: Thank you for raising this important point. In our systematic review and meta-analysis protocol on PND in individuals with a history of traumatic brain injury, we will include race, education, income, and health insurance status if these variables are reported in the included studies. We recognize that socioeconomic factors, such as lower education, lower income, and access to health insurance, can influence cognitive health outcomes and potentially contribute to variability in the prevalence or severity of PND in this population.

Line 162-166: We will collect data such as first authors, year of publication, patient age, gender, BMI, time of anesthesia, type of anesthesia, duration of anesthesia race, education, income, and health insurance status, country, duration of post-operative stay, time of cognitive assessment after surgery and anesthesia, return to work, comorbidity, diagnosis, and exposure.”

Line 205-210: We acknowledge that not all studies may report these variables, particularly socioeconomic factors and healthcare access (such as health insurance coverage). If such data are available, we will incorporate them in subgroup analyses or meta-regression models to explore their potential effects. In cases where data are insufficient or inconsistently reported, we will highlight this limitation and discuss its implications in the final manuscript.

Reviewer 3 Report

Comments and Suggestions for Authors

This topic is interesting, look at these points to improve it:

- Lines 13-24: It is unclear why the authors wrote about neurocognitive changes after general anesthesia and surgery, but the topic is neurocognitive changes after traumatic brain injury. Improve this part. - It is not clear what the inclusion criteria for this protocol are. - Lines 244-251. Improve this part "The implications of our research can be significant for..." the authors should better explain what they think the final outcome of the review will be

Author Response

 Comment #1: - This topic is interesting, look at these points to improve it: Lines 13-24: It is unclear why the authors wrote about neurocognitive changes after general anesthesia and surgery, but the topic is neurocognitive changes after traumatic brain injury. Improve this part. - It is not clear what the inclusion criteria for this protocol are. –

Response: Thank you for finding the topic interesting. We have edited the text to clarify that the main topic of this study is to analyze whether patients with a history of mild to moderate TBI have an increased risk of developing PND after surgery under GA.

Lines 13-21: “Preexisting neurodegenerative changes, stress and inflammation that become more common with advanced age may contribute to accelerated neurocognitive decline after general anesthesia (GA) and surgery, a condition known as postoperative neurocognitive disorder (PND). PND affects millions of senior patients every year. Whether specific groups of young adults with pathophysiological conditions that involve dysregulated stress response systems, neuroinflammation, neurodegenerative and neurocognitive abnormalities, such as traumatic brain injury (TBI), are also more susceptible to PND is not known. We will analyze published research data to determine whether patients with a history of mild to moderate TBI have an increased risk of developing PND after surgery under GA.”

Comment #2: Lines 244-251. Improve this part "The implications of our research can be significant for..." the authors should better explain what they think the final outcome of the review will be

Response: We have edited the text to improve its clarity

Lines 262-268: “The results of our study may contribute to the improvement of patient care by identifying factors that predispose patients with a history of TBI and extracranial surgery, as well as their unexposed offspring, to PND. These results may offer insights for optimizing anesthesia regimens, timing of surgeries, and perioperative care protocols to minimize the risk of PND. The proposed study may also help identify neurocognitive and molecular biomarkers of PND in patients with a history of TBI and their offspring, and potential translational pharmacotherapeutic approaches to minimize the consequences of PND in such patients. Finally, the proposed study may provide evidence for the importance of considering the neurocognitive consequences of parental PND for the offspring of patients with a history of TBI.” 

Round 2

Reviewer 3 Report

Comments and Suggestions for Authors

Good